# TEXT-GUIDED VISUAL PROMPT TUNING FOR VISION-LANGUAGE MODELS

## ABSTRACT

Prompt tuning has become a crucial technique for adapting pre-trained vision-language models (VLMs) to various downstream tasks. Recent advancements introduce multi-modal learnable prompts to enhance the creation of task-specific classifiers. Despite their utility, these methods commonly encounter challenges in generalizing to unseen classes, as their symmetrically designed visual prompt struggles to capture task-relevant textual knowledge and lacks the flexibility in adjusting to novel test class distributions. To tackle these obstacles, we propose a novel Text-Guided Visual Prompt Tuning (TGVP) method, which uniquely leverages the robust generalizability of textual knowledge to guide the generation of visual prompt. Our method introduces a simple yet effective Text-Knowledge Guidance Module that dynamically incorporates visual prompt with task-relevant textual knowledge through cross-attention mechanism. The generated text-guided visual prompt endows the visual encoder with semantic awareness and thus enhances both generalization and discriminability of VLMs across various scenarios. Comprehensive experiments demonstrate that TGVP significantly outperforms existing methods in base-to-novel generalization, cross-dataset transfer, and domain generalization tasks, offering a substantial improvement in VLM adaptation.

## 1 INTRODUCTION

Foundational vision-language models (VLMs), such as CLIP (Radford et al., 2021b) and BLIP (Li et al., 2022a), pre-trained on large-scale image-text pairs, have demonstrated remarkable generalization abilities across diverse downstream vision tasks. However, training models from scratch generally requires large labeled datasets, which limits their applicability to downstream tasks with fewer samples. To overcome this, parameter-efficient adaptation techniques such as prompt tuning (Zhou et al., 2022c), adapters (He et al., 2021), and LoRA (Hu et al., 2021) have been introduced. Among these, prompt tuning has become a prominent approach for maximizing the potential of VLMs, balancing parameter efficiency while effectively preserving pre-trained knowledge.

CoOp (Zhou et al., 2022c) was the pioneer in introducing prompt tuning by concatenating learnable contextual tokens to class names, demonstrating its effectiveness. However, these learnable prompts face over-fitting problem, particularly with limited training data, leading to degraded generalization on unseen classes. To mitigate the issue, subsequent methods have incorporated various regularization techniques when updating these prompts. For instance, CoCoOp (Zhou et al., 2022b) conditions on image features to enhance the learnable textual prompts with instance-level visual information. Follow-up methods such as KgCoOp (Yao et al., 2023b), ProGrad (Zhu et al., 2023a), and ProReg (Zhu et al., 2023b) impose stronger constraints on learnable prompts from vanilla CLIP features to ensure that they effectively encapsulate essential general knowledge.

Alternative methods, such as PromptSRC (Khattak et al., 2023c), address the over-fitting problem by enhancing cross-modal alignment in the shared vision-language space. Instead of solely updating textual prompts, they take advantage of the multi-modal encoding capabilities of VLMs by learning both textual and visual prompts. While these two-branch designs yield improved results, both types of prompts are independently optimized to over-fit the base classes and cannot effectively handle novel classes. Additionally, the lack of cross-modal interactions constrains the sufficient multi-modal alignment.

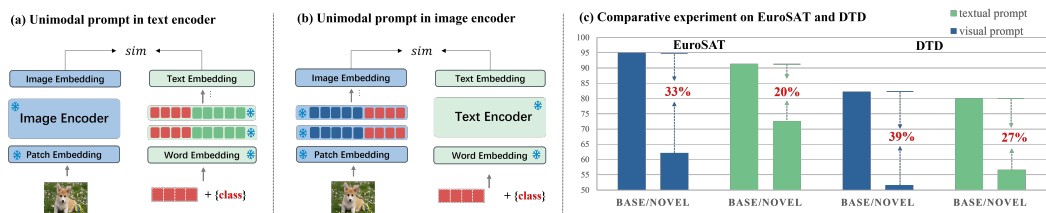

Figure 1: Prompt tuning in different modalities. (a) Unimodal prompt in text encoder. (b) Unimodal prompt in image encoder. (c) Experiment shows that textual prompts result in less performance degradation on novel classes compared to visual prompt.

A natural question arises: which modality of prompt tuning enhances the model's generalization ability? To gain a deeper understanding of prompt learning across different modalities, we set up a comparative experiment in which only one modality of prompts (either visual or text) is set as learnable and trained on base classes. Figure 1 illustrates that textual prompts result in less performance degradation on novel classes compared to visual prompts, exhibiting stronger generalization capability. This motivates us to incorporate textual knowledge into visual branch for enhanced the generalization capability when adapting VLMs. The previous method, MaPLee (Khattak et al., 2023a), also notices this problem and establishes a mapping from textual prompts to visual prompts for better alignment of two modalities. However, it has certain limitations. First, the source of textual information is confined to fixed text prompts, which are uniform across both seen and unseen scenarios, thereby hindering effective adaptation to unseen classes.Moreover, the simple symmetrically projection mechanism is insufficient for information interaction between visual and textual modalities, as textual features naturally contain semantic information while visual features carry local patch information from the current image.

Therefore, to mitigate these limitations, we propose a novel Text-Guided Visual Prompt Tuning (TGVP) approach to leverage the generalization capability of textual knowledge to guide the generation of visual prompts, focusing on both the knowledge source and the method of knowledge transfer. We emphasize that **high-level textual semantics are key to facilitating the learning of generalizable visual prompts**. Instead of using textual prompts to enhance their visual counterparts, we utilize the text embeddings, which encode high-level semantic information, as the knowledge source to guide the optimization of visual prompts. To better harness this knowledge, we propose a novel Text-Knowledge Guidance Module to dynamically select and fuse the task-relevant textual knowledge into the visual encoder, enabling it to be semantically aware of both seen and unseen classes. Specifically, both visual prompt tokens and the **CLS** token cooperatively serve as queries to dynamically select the most relevant textual guidance through cross-attention mechanism. We conducted extensive experiments to evaluate the proposed TGVP approach on base-to-novel generalization, cross-dataset transfer, few-shot classification, and domain generalization tasks. Experimental results demonstrate the significantly superior performance of TGVP compared to existing state-of-the-art methods. The main contributions can be summarized as follows:

- We point out that text embeddings can be leveraged as knowledge source at the cross-modal interaction rather than text prompts, thereby enhancing both discriminability and generalizability of the visual representation in VLMs.
- We propose a novel Text-Guided Visual Prompt Tuning mechanism, which dynamically transfers textual knowledge to guide the generation of visual prompt, making it semantically aware for both seen and unseen classes.
- Extensive and comprehensive experiments have validated the consistent effectiveness and the superior performance by significant margins.

## 2 RELATED WORK

### 2.1 VISION-LANGUAGE MODELS

In recent years, Vision-Language Models (VLMs) (Radford et al., 2021b; Jia et al., 2021; Yuan et al., 2021; Li et al., 2022a) have emerged as a powerful paradigm, effectively leveraging visual

and textual modalities trained on large-scale image-text datasets. Current research underscores that these models, pre-trained on extensive image-text pairs sourced from the internet, possess the capability to comprehend the semantics of images in conjunction with their corresponding textual descriptions (Radford et al., 2021b; Yu et al., 2022). Notably, recent studies (Zhang et al., 2021; Zhou et al., 2022c) have demonstrated that, with a deep understanding of open-vocabulary concepts, VLMs exhibit proficiency in addressing a diverse array of downstream visual tasks, including but not limited to image retrieval (Duan et al., 2022), depth estimation (Hu et al., 2023), visual grounding (Li et al., 2022b), and visual question answering (Duan et al., 2022).

## 2.2 PROMPT TUNING

Prompt tuning (Gan et al., 2022; Ouali et al., 2023; Lee et al., 2023; Zang et al., 2022; Radford et al., 2021a) has emerged as a prominent approach for adapting pre-trained VLMs to downstream tasks by leveraging learnable tokens to encapsulate task-specific knowledge. In models like CLIP, handcrafted templates such as "a photo of a [CLASS]" are employed to encode textual embeddings for zero-shot predictions. However, these handcrafted prompts often fall short of capturing the subtle nuances required for downstream tasks. To overcome this limitation, textual prompt tuning techniques have been developed to enhance textual embeddings by inferring a set of learnable tokens combined with class tokens.

For example, CoOp (Zhou et al., 2022c) replaces static handcrafted prompts with dynamic, learnable soft prompts. To further improve the generalization capability of these learnable prompts, CoCoOp (Zhou et al., 2022b) introduces image-conditional prompts that integrate image features with learnable tokens. Additionally, approaches like KgCoOp (Yao et al., 2023b), ProGrad (Zhu et al., 2023a) impose constraints on learnable prompts to ensure they encapsulate essential, generalized knowledge. Beyond textual prompt tuning, recent advancements such as MaPLe (Khattak et al., 2023a) and PromptSRC (Khattak et al., 2023c) propose joint optimization of both visual and textual prompts. CLIP-Adapter (Gao et al., 2021) integrates an adapter mechanism to fine-tune both visual and textual embeddings, further enhancing model adaptability.

However, previous unimodal prompt tuning methods often struggle with generalization to unseen classes due to inadequate modeling of test class distributions, while existing multi-modal prompt tuning strategies are also hampered by limited cross-modality information exchange, restricting performance improvements. To tackle these obstacles, we propose Text-Guided Visual Prompt Tuning (TGVP), which transfers general textual knowledge into the vision encoder via a Text-Knowledge Guidance (TKG) Module. Utilizing a streamlined cross-attention mechanism, visual prompts, alongside the CLS token, dynamically select relevant textual guidance. This process enables the creation of a semantic-aware vision classifier that effectively adapts to diverse downstream tasks.

## 3 METHOD

### 3.1 PRELIMINARIES

**CLIP** CLIP is a representative and powerful Vision-Language Pre-Trained Model (VL-PTM) that includes a vision encoder $V$ and a text encoder $T$, both of which are well-mapped to a common feature space for alignment.

Given an input image $x$, the vision encoder extracts its representation, denoted as $I_x = V(x)$. For each downstream dataset with $k$ classes, a manual prompt template like "a photo of <CLASS>" is used. The text encoder generates feature representations for each class. During training, CLIP maximizes the cosine similarity between matched image and text representations while minimizing it for unmatched pairs. In zero-shot inference, the prediction probability for the $i$-th class is:

$$p(y = i \mid I) = \frac{\exp(\cos(I_x, T(y_i))/\tau)}{\sum_{j=1}^{k} \exp(\cos(I_x, T(y_j))/\tau)}, \tag{1}$$

where $\tau$ is a temperature parameter, and $\cos$ represents cosine similarity.

**Prompt Engineering** To further improve the discriminative capabilities of VLMs, CoOp (Context Optimization) introduces learnable tokens into the prompt templates. Rather than employing a static

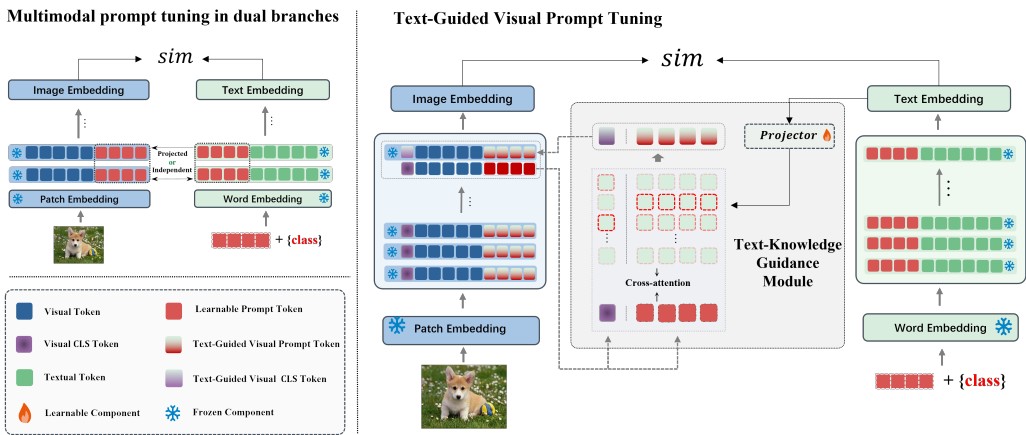

Figure 2: An overview of previous multi-modal prompt tuning methods and our proposed Text-Guided Prompt Tuning (TGVP), which dynamically transfers textual knowledge to guide the generation of visual prompt through a novel Text-Knowledge Guidance Module.

prompt like "a photo of <CLASS>", CoOp replaces it with a series of learnable tokens. The CoOp prompt template is defined as follows:

$$z_i = [T_1][T_2]\dots[T_n][<CLASS>], \tag{2}$$

where $[T_1], [T_2], \dots, [T_n]$ represent learnable tokens. These tokens are optimized during training to enhance the alignment of text and image representations.

### 3.1.1 DEEP LANGUAGE PROMPTING.

To further enhance the learning of language context prompts, some methods introduce $n$ learnable tokens $\{P^i \in \mathbb{R}^{d_t}\}_{i=1}^q$ in the language branch of the CLIP model. The input embeddings now follow the structure $[P^1, P^2, \cdots, P^q, X_0]$, where $X_0$ denotes the fixed input tokens. New learnable tokens are additionally introduced within each transformer block of the text encoder $(\mathcal{T}_i)$ up to a specified depth $L$:

$$[\_, X_i] = \mathcal{T}_i([P_{i-1}, X_{i-1}]) \quad i = 1, 2, \dots, L. \tag{3}$$

Here, $[\cdot, \cdot]$ indicates the concatenation operation and $[\_]$ denotes the tokens to be replaced by the prompt in next layer.

### 3.1.2 DEEP VISION PROMPTING.

Similarly, in the vision branch of CLIP, we introduce $q$ learnable tokens $\{\tilde{P}^i \in \mathbb{R}^{d_v}\}_{i=1}^q$, which are integrated with the input image tokens. Additional learnable tokens are incorporated into deeper transformer layers of the image encoder $(\mathcal{V})$ up to a depth $L$:

$$[c_i, E_i, \_] = \mathcal{V}_i([c_{i-1}, E_{i-1}, \tilde{P}_{i-1}]) \quad i = 1, 2, \dots, L, \tag{4}$$

where $c_i$ is the CLS token in the $i$-th layer, $E_i$ denotes fixed input tokens.

### 3.2 TEXT-GUIDED VISUAL PROMPT TUNING

**Overview.** In Figure 2, we present an overview of our proposed TGVP method, alongside a brief comparison with existing multi-modal prompt techniques. As depicted in Figure 2(a), prior methods tend to design visual prompts symmetrically to textual prompts, treating them as direct counterparts or projections. However, this approach limits the extent to which visual prompts can internalize textual knowledge, thus preventing them from serving as optimal context for target classes at a natural semantic level. This restricted interaction between modalities further hinders model performance enhancement. To address these challenges, we introduce TGVP, which uniquely harnesses the robust generalizability of textual knowledge to guide the generation of visual prompt. As illustrated in Figure 2, TGVP employs a Text-Knowledge Guidance (TKG) Module that first transfers general

textual embeddings into targeted guidance for the visual modality. Within the visual feature space, visual prompt tokens and the corresponding CLS token act as queries, dynamically selecting the most relevant text guidance via a streamlined cross-attention mechanism. This selected guidance, combined with the original visual prompt tokens and CLS token, enables the creation of a semantic-aware vision classifier, capable of dynamically perceiving relevant textual knowledge and adapting effectively to unseen classes.

**Text-Knowledge Guidance Module.** Given the text embedding $W_{text} \in \mathbb{R}^{N_c \times D_t}$ with $N_c$ classes generated from text encoder $\mathcal{T}$, the TKG Module is designed to enable the visual branch to incorporate guidance from the text embedding $W_{text}$, where $N_c$ represents the number of text categories, $L_{dvp}$ denotes the length of text guidance tokens/visual prompts, and $D_v$ is the dimensionality of the vision embedding space. As shown in Figure 2, text knowledge, represented by textual embedding $W_{text}$ is first projected into vision embedding space as $W_{guide} \in \mathbb{R}^{N_c \times D_v \times L_{dvp}}$. The projector is realized by a down-project layer $W_{down} \in \mathbb{R}^{D_t \times D_{mid}}$ followed by an up-project layer $W_{up} \in \mathbb{R}^{D_{mid} \times D^{'}}$, where $\mathbf{D}^{'} = D_v \times L_{dvp}$.

$$W_{guide} = \text{Projector}(W_{text}) \tag{5}$$

After obtaining the text knowledge guidance $W_{guide}$, a simplified cross-attention block is proposed to capture the most relevant text category knowledge as guidance for current vision task at various levels. More specifically, we conceptualize and implement a streamlined cross-attention module designed to facilitate the alignment and feature integration between textual and visual prompts.

Initially, we compute the dot-product similarity between the text guidance tokens and the visual prompt, represented by $\mathbf{P} \in \mathbb{R}^{L_{dvp} \times D_v}$, to obtain the similarity matrix $\mathbf{S}$. And for each visual prompt token, we identify the top-k most relevant text categories. Let $\mathbf{S}_j$ denote the similarity values corresponding to the $j$-th visual prompt token:

$$\mathbf{S} = \mathbf{W}_{guide}\mathbf{P}^{\top} \tag{6}$$

$$\mathbf{T}_j^{topk} = \mathbf{TopK}(\mathbf{S}_j, k), j \in [1, L_{dvp}] \tag{7}$$

Here, $\mathbf{T}_j^{topk}$ encapsulates the top-k most relevant text categories corresponding to the $j$-th visual prompt.

After the selection of top-k text categories, we employ a softmax function with a temperature coefficient $\tau$ to modulate the distribution, yielding the attention map:

$$\mathbf{A}_j^{topk} = \text{softmax}\left(\frac{\mathbf{S}_j^{topk}}{\tau}\right) \tag{8}$$

Then we proceed to perform feature aggregation for the top-k text categories based on the derived attention map. For each visual prompt token, the corresponding text guidance is accomplished by computing a weighted sum of the top-k text category knowledge, where the attention scores function as weights:

$$\mathbf{T}_j^{guide} = \sum_{i=1}^{k} \alpha_{ij}\mathbf{T}_{ij}^{topk} \tag{9}$$

In this equation, $\alpha_{ij}$ represents the attention weight from the map $\mathbf{A}_j^{top}$, and $\mathbf{T}_{ij}^{top}$ corresponds to the $i$-th top-k text category knowledge.

Finally, the text-guided visual prompt can be formulated as the incorporation of the textual guidance with the visual prompts through a linear combination utilizing an Exponential Moving Average (EMA) approach. Specifically, for each text-guided visual prompt token, the final representation is computed as follows:

$$\mathbf{P}_j^{tg} = \lambda\mathbf{T}_j^{guide} + (1 - \lambda)\mathbf{P}_j \tag{10}$$

where $\lambda$ is the EMA coefficient that controls the balance between the newly fused textual guidance token and the original visual prompt token. The EMA-based linear fusion ensures that the final

representation retains essential visual characteristics while integrating the most relevant textual guidance, thereby enhancing overall representation quality.

Given that the **CLS** token encapsulates rich semantic information about the current image, we can further enhance the representation capabilities of the vision classifier by utilizing the **CLS** token as a query. By applying the same aforementioned steps, we derive an instance-level text-guided **CLS** token, referred to as $\mathbf{C}^{tg}$.

Assuming we insert the text-guided visual prompt $\mathbf{P}^{tg}$ into the $l$-th transformer layer of the Image Encoder $\mathbf{\Theta_l}$, the prompted visual feature $\mathbf{F}_l$ is presented as

$$\mathbf{F}_l = \mathbf{\Theta}_l([\mathbf{C}^{tg}_{l-1}, \mathbf{E}_{l-1}, \mathbf{P}^{tg}_{l-1}]) \tag{11}$$

where $\mathbf{C}^{tg}_{l-1}$ is the text-guided **CLS** token, $\mathbf{P}^{tg}_{l-1}$ is text-guided visual prompt and $\mathbf{E}_{l-1}$ is the rest of vision tokens.

## 4 EXPERIMENTS

In this section, we present quantitative results of our method and comprehensive comparisons with other state-of-the-art methods to demonstrate the effectiveness of our proposed TGVP. Similar to previous works, we evaluate the proposed TGVP across four challenging task settings:

- **Base-to-Novel Generalization.** We evaluate the generalization ability of our approach in a zero-shot context by dividing the datasets into base and novel classes. The model is trained with a few examples from the base classes and then tested on unseen novel classes to assess its performance.

- **Cross-Dataset Transfer.** To examine the transferability of our method, we conduct a direct evaluation of our ImageNet-trained model across a diverse array of external datasets. Adhering to established protocols, the model is trained on all 1,000 ImageNet classes under a few-shot paradigm.

- **Domain Generalization.** We further test the robustness of our method by evaluating it on out-of-distribution (OOD) datasets. Specifically, the model trained on ImageNet is assessed on four different ImageNet variants, each representing a different type of domain shift.

- **Few-shot Classification.** This scenario allows us to compare the model's learning capacity under very limited supervision. It also helps determine whether our approach effectively learns both task-specific and generalizable knowledge.

**Datasets.** For base-to-novel generalization, cross-dataset transfer tasks, we follow previous work (Radford et al., 2021b; Zhou et al., 2022c;b) to conduct the experiments on 11 representative image classification datasets, including ImageNet (Deng et al., 2009) and Caltech101 (Fei-Fei et al., 2004) for generic object classification; OxfordPets (Parkhi et al., 2012), StanfordCars (Krause et al., 2013), Flowers102 (Nilsback & Zisserman, 2008), Food101 (Bossard et al., 2014), and FGV-CAircraft (Maji et al., 2013) for fine-grained classification; SUN397 (Xiao et al., 2010) for scene recognition; UCF101 (Soomro et al., 2012) for action recognition; DTD (Cimpoi et al., 2014) for texture classification; and EuroSAT (Helber et al., 2019) for satellite image recognition. For domain generalization, we utilize ImageNet as the source dataset and four ImageNet variants as target datasets including ImageNet-A (Hendrycks et al., 2021b), ImageNet-R (Hendrycks et al., 2021a), ImageNet-V2 (Recht et al., 2019), ImageNet-Sketch (Wang et al., 2019).

**Baselines.** The baselines used for comparison in the experimental section include: CLIP (Radford et al., 2021b), CoOp Zhou et al. (2022c), CoCoOp Zhou et al. (2022a), ProGrad Zhu et al. (2023a), WiSE-FT Wortsman et al. (2022), KgCoOp Yao et al. (2023a), PromptSRC Khattak et al. (2023b), MaPLe Khattak et al. (2023a), TCP Yao et al. (2024), DAPT Cho et al. (2023)

**Implementation Details.** For a fair comparison, all experiments are conducted based on the CLIP with the backbone of ViT-B/16 (Dosovitskiy et al., 2021) and reported results are averaged over 3 runs. We employ deep prompting, and the prompts length in both text/vision branch, denoted by $L$, is set as 4 with a normal distribution. The SGD optimizer is adopted for optimization with the learning

Table 1: Comparison with state-of-the-art methods on base-to-novel generalization using the ViT-B/16 backbone. Our proposed approach exhibits superior generalization performance across eleven recognition datasets, surpassing existing methods. The highest-performing results are highlighted in **bold**, while the second-best outcomes are underlined. HM indicates the harmonic mean.

(a) **Average over 11 datasets.**

| Method | Base | Novel | HM |
|---|---|---|---|
| CLIP | 69.34 | 74.22 | 71.70 |
| CoOp | 82.69 | 63.22 | 71.66 |
| CoCoOp | 80.47 | 71.69 | 75.83 |
| KgCoOp | 80.73 | 73.60 | 77.00 |
| MaPLe | 82.28 | 75.14 | 78.55 |
| TCP | 84.13 | 75.36 | 79.51 |
| PSRC | 84.26 | 76.10 | 79.97 |
| Ours | **85.10** | **77.73** | **81.24** |
| | +0.84 | +1.63 | +1.27 |

(b) ImageNet.

| Method | Base | Novel | HM |
|---|---|---|---|
| CLIP | 72.43 | 68.14 | 70.22 |
| CoOp | 76.47 | 67.88 | 71.92 |
| CoCoOp | 75.98 | 70.43 | 73.10 |
| KgCoOp | 75.83 | 69.96 | 72.78 |
| MaPLe | 76.66 | 70.54 | 73.47 |
| TCP | 77.27 | 69.87 | 73.38 |
| PSRC | 77.60 | 70.73 | 74.01 |
| Ours | **77.74** | **70.83** | **74.12** |
| | +0.14 | +0.10 | +0.11 |

(c) Caltech101.

| Method | Base | Novel | HM |
|---|---|---|---|
| CLIP | 96.84 | 94.00 | 95.40 |
| CoOp | 98.00 | 89.81 | 93.73 |
| CoCoOp | 97.96 | 93.81 | 95.84 |
| KgCoOp | 97.72 | 94.39 | 96.03 |
| MaPLe | 97.74 | 94.36 | 96.02 |
| TCP | 98.23 | 94.67 | 96.42 |
| PSRC | 98.10 | 94.03 | 96.02 |
| Ours | **98.55** | **94.72** | **96.57** |
| | +0.32 | +0.05 | +0.15 |

(d) OxfordPets.

| Method | Base | Novel | HM |
|---|---|---|---|
| CLIP | 91.17 | 97.26 | 94.12 |
| CoOp | 93.67 | 95.29 | 94.47 |
| CoCoOp | 95.20 | 97.69 | 96.43 |
| KgCoOp | 94.65 | 97.76 | 96.18 |
| MaPLe | 94.67 | 97.20 | 95.92 |
| TCP | 95.43 | 97.76 | 96.58 |
| PSRC | 95.33 | 97.30 | 96.30 |
| Ours | **96.36** | **98.03** | **97.18** |
| | +0.93 | +0.27 | +0.60 |

(e) StanfordCars.

| Method | Base | Novel | HM |
|---|---|---|---|
| CLIP | 63.37 | 74.89 | 68.65 |
| CoOp | 78.12 | 60.40 | 68.13 |
| CoCoOp | 70.49 | 73.59 | 72.01 |
| KgCoOp | 71.76 | 75.04 | 73.36 |
| MaPLe | 72.94 | 74.00 | 73.47 |
| TCP | 80.80 | 74.13 | 77.32 |
| PSRC | 78.27 | 74.97 | 76.58 |
| Ours | **80.86** | **75.45** | **78.06** |
| | +0.06 | +0.41 | +0.74 |

(f) Flowers.

| Method | Base | Novel | HM |
|---|---|---|---|
| CLIP | 72.08 | **77.80** | 74.83 |
| CoOp | 97.60 | 59.67 | 74.06 |
| CoCoOp | 94.87 | 71.75 | 81.71 |
| KgCoOp | 95.00 | 74.73 | 83.65 |
| MaPLe | 95.92 | 72.46 | 82.56 |
| TCP | 97.73 | 75.57 | 85.23 |
| PSRC | 98.07 | 76.50 | 85.95 |
| Ours | **98.27** | 76.86 | **86.25** |
| | +0.20 | -0.94 | +0.30 |

(g) Food101.

| Method | Base | Novel | HM |
|---|---|---|---|
| CLIP | 90.10 | 91.22 | 90.66 |
| CoOp | 88.33 | 82.26 | 85.19 |
| CoCoOp | 90.70 | 91.29 | 90.99 |
| KgCoOp | 90.05 | 91.70 | 91.09 |
| MaPLe | 90.71 | 92.05 | 91.38 |
| TCP | 90.57 | 91.37 | 90.97 |
| PSRC | 90.67 | 91.53 | 91.10 |
| Ours | **90.88** | **92.28** | **91.57** |
| | +0.17 | +0.23 | +0.19 |

(h) FGVCAircraft.

| Method | Base | Novel | HM |
|---|---|---|---|
| CLIP | 27.19 | 36.29 | 31.09 |
| CoOp | 40.44 | 22.30 | 28.75 |
| CoCoOp | 33.41 | 23.71 | 27.74 |
| KgCoOp | 36.21 | 33.55 | 34.83 |
| MaPLe | 37.44 | 35.61 | 36.50 |
| TCP | 41.97 | 34.43 | 37.83 |
| PSRC | 42.73 | 37.87 | 40.15 |
| Ours | **43.27** | **38.65** | **40.83** |
| | +0.54 | +0.78 | +0.68 |

(i) DTD.

| Method | Base | Novel | HM |
|---|---|---|---|
| CLIP | 53.24 | 59.90 | 56.37 |
| CoOp | 79.44 | 41.18 | 54.24 |
| CoCoOp | 77.01 | 56.00 | 64.85 |
| KgCoOp | 77.55 | 54.99 | 64.35 |
| MaPLe | 82.77 | 58.07 | 68.25 |
| TCP | 80.36 | 59.18 | 68.16 |
| PSRC | 83.37 | 62.97 | 71.75 |
| Ours | **83.62** | **64.37** | **72.43** |
| | +0.25 | +1.40 | +0.68 |

(j) SUN397.

| Method | Base | Novel | HM |
|---|---|---|---|
| CLIP | 69.36 | 75.35 | 72.23 |
| CoOp | 80.60 | 65.89 | 72.51 |
| CoCoOp | 79.74 | 76.86 | 78.27 |
| KgCoOp | 80.29 | 76.53 | 78.36 |
| MaPLe | 80.82 | 78.70 | 79.75 |
| TCP | 82.63 | 78.20 | 80.35 |
| PSRC | 82.67 | 78.47 | 80.52 |
| Ours | **82.88** | **78.85** | **80.81** |
| | +0.21 | +0.15 | +0.29 |

(k) EuroSAT.

| Method | Base | Novel | HM |
|---|---|---|---|
| CLIP | 56.48 | 64.05 | 60.03 |
| CoOp | 92.19 | 54.74 | 68.69 |
| CoCoOp | 87.49 | 60.04 | 71.21 |
| KgCoOp | 85.64 | 64.34 | 73.48 |
| MaPLe | 94.07 | 73.23 | 82.35 |
| TCP | 91.63 | 74.73 | 82.32 |
| PSRC | 92.90 | 73.90 | 82.32 |
| Ours | **95.88** | **85.42** | **90.35** |
| | +1.81 | +10.69 | +8.00 |

(l) UCF101.

| Method | Base | Novel | HM |
|---|---|---|---|
| CLIP | 70.53 | 77.50 | 73.85 |
| CoOp | 84.69 | 56.05 | 67.46 |
| CoCoOp | 82.33 | 73.45 | 77.64 |
| KgCoOp | 82.89 | 76.67 | 79.65 |
| MaPLe | 83.00 | 78.66 | 80.77 |
| TCP | 87.13 | **80.77** | **83.83** |
| PSRC | 87.10 | 78.80 | 82.74 |
| Ours | **87.83** | 79.67 | 83.55 |
| | +0.73 | -1.10 | -0.28 |

rate of 2e-3 and the batch size of 4, and the training epochs is 20. All experiments are conducted on a single A800-40G GPU.

## 4.1 BASE TO NOVEL GENERALIZATION

As illustrated in Table 1, the proposed TGVP method achieves the highest average performance in terms of Base/Novel/HM, specifically obtaining 85.10%, 77.73%, and 81.24%. Since TGVP can

dynamically leverage the beneficial textual knowledge to generate semantic-aware visual classifier when confronting unseen classes, it obtains the best performance on Novel classes of 77.73%, achieving a 1.63% improvement over the existing state-of-the-art, PromptSRC. The superior performance of TGVP verifies the necessity and the significance of using textual knowledge to guide the generation of visual prompt. Additionally, the TGVP also demonstrates strong few-shot performance on Base classes with a 0.97% improvement over the PSRC. In conclusion, the superior performance of TGVP demonstrates that the text-guided semantic-aware visual prompts can enhance both discriminative and generalization capacities of VLMs.

Table 2: Comparison of our method with existing approaches on cross-dataset evaluation. Overall, our method demonstrates superior generalization capabilities with the highest average accuracy on 11 datasets.

| | Source | Target | | | | | | | | | | |
|---|---|---|---|---|---|---|---|---|---|---|---|---|
| | ImageNet | Caltech101 | OxfordPets | Cars | Flowers | Food101 | Aircraft | SUN397 | DTD | EuroSAT | UCF101 | Avg. |
| CoOp | 71.51 | 93.70 | 89.14 | 64.51 | 68.71 | 85.30 | 18.47 | 64.15 | 41.92 | 46.39 | 66.55 | 63.88 |
| CoCoOp | 71.02 | 94.43 | 90.14 | 65.32 | 71.88 | 86.06 | 22.94 | 67.36 | 45.73 | 45.37 | 68.21 | 65.74 |
| MaPLe | 70.72 | 93.53 | 90.49 | 65.57 | 72.23 | 86.20 | 24.74 | 67.01 | 46.49 | 48.06 | 68.69 | 66.30 |
| TCP | 71.40 | 93.97 | 91.25 | 64.69 | 71.21 | 86.69 | 23.45 | 67.15 | 44.35 | 51.45 | 68.73 | 66.29 |
| PSRC | 71.27 | 93.60 | 90.25 | 65.70 | 70.25 | 86.15 | 23.90 | 67.10 | 46.87 | 45.50 | 68.75 | 65.81 |
| **Ours** | **71.88** | **95.42** | **91.44** | **65.97** | **73.45** | **87.18** | **26.26** | **68.04** | **47.96** | **52.45** | **70.21** | **67.83** |

Table 3: Comparison of our method with existing approaches on few-shot learning with 4-shot samples. Overall, our method demonstrates superior discriminative capacity with the highest average accuracy on 11 datasets.

| Method | ImageNet | Caltech101 | OxfordPets | Cars | Flowers | Food101 | Aircraft | SUN397 | DTD | EuroSAT | UCF101 | Avg. |
|---|---|---|---|---|---|---|---|---|---|---|---|---|
| CLIP | 66.70 | 93.30 | 89.10 | 65.70 | 70.70 | 85.90 | 24.90 | 62.60 | 44.30 | 48.30 | 67.60 | 65.37 |
| CoOp | 69.37 | 94.44 | 91.30 | 72.73 | 91.14 | 82.58 | 33.18 | 70.13 | 58.57 | 68.62 | 77.41 | 73.59 |
| CoCoOp | 70.55 | 94.98 | 93.01 | 69.10 | 82.56 | 86.64 | 30.87 | 70.50 | 54.79 | 63.83 | 74.99 | 71.98 |
| ProGrad | 70.21 | 94.93 | 93.21 | 71.75 | 89.98 | 85.77 | 32.93 | 71.17 | 57.72 | 70.84 | 77.82 | 74.21 |
| KgCoOp | 70.19 | 94.65 | 93.20 | 71.98 | 90.69 | 86.59 | 32.47 | 71.79 | 58.31 | 71.06 | 78.40 | 74.48 |
| MaPLe | 70.67 | 94.30 | 92.05 | 68.70 | 80.80 | 86.90 | 29.03 | 71.47 | 54.73 | 54.87 | 73.70 | 70.66 |
| DAPT | 70.80 | 94.23 | 92.17 | 74.40 | 92.37 | 83.60 | 32.47 | 72.20 | 61.37 | 72.73 | 79.40 | 75.07 |
| PSRC | 70.80 | 94.77 | 93.23 | 71.83 | 91.31 | 86.06 | 22.80 | 72.80 | 60.64 | 75.02 | 79.35 | 75.33 |
| TCP | 70.48 | 95.00 | 91.90 | 76.30 | **94.40** | 85.30 | 36.20 | 72.11 | 63.97 | 77.43 | 80.83 | 76.72 |
| **Ours** | **71.28** | **95.25** | **93.45** | **76.56** | 93.42 | **87.12** | **36.87** | **74.14** | **65.43** | **86.42** | **80.83** | **78.21** |

## 4.2 CROSS-DATASET TRANSFER

Unlike base-to-new generalization, cross-dataset transfer presents a more rigorous challenge in generalization than base-to-novel transfer, as it demands adaptation across distinct datasets rather than within a single one. Comparative results with CoOp, CoCoOp, MaPLe, TCP and PSRC are presented in Table 2. Our TGVP method consistently demonstrates superior performance on both the source and target datasets, achieving a target average of 67.83%, and surpassing existing state-of-the-art, MaPLE, by 1.53%. The superior performance highlights our method's robust dynamic representation capabilities for unseen data.

## 4.3 FEW-SHOT CLASSIFICATION

To further validate the strong representation ability with fewer limitation of the proposed TGVP, we perform few-shot classification across all 11 datasets using K-shot labeled source images. The evaluation is conducted on the standard testing domain, which shares the same class space as the training classes. A comparison of the 4-shot setting between the proposed TGVP and existing methods is presented in Table 3. The results indicate that our method consistently outperforms existing approaches, achieving an average performance of 78.21%, which represents a 1.39% gain over the previous state-of-art, TCP. Notably, our method outperforms TCP by 8.99% on EuroSAT, a satellite image dataset fundamentally distinct from ImageNet, showcasing our method's powerful visual representation capabilities in few-shot scenarios.

## 4.4 DOMAIN GENERALIZATION

Table 4 presents a summary of the performance of our TGVP in comparison to previous methods on out-of-distribution datasets. We evaluated our model, which is trained on ImageNet, directly on these target datasets. TGVP consistently outperforms all other approaches, achieving the highest average accuracy of 61.07%. These results suggest that our TGVP effectively enhances generalization for datasets exhibiting domain shifts.

Table 4: Comparison with other methods on robustness (%) to domain generalization.

| | Source | Target | | | | |
|---|---|---|---|---|---|---|
| | ImageNet | -V2 | -Sketch | -A | -R | Avg. |
| CLIP | 66.73 | 60.83 | 46.15 | 47.77 | 73.96 | 57.18 |
| WiSE-FT | **73.02** | **65.19** | 49.09 | 49.81 | 77.63 | 60.43 |
| CoOp | 71.51 | 64.20 | 47.99 | 49.71 | 75.21 | 59.28 |
| CoCoOp | 71.02 | 64.07 | 48.75 | 50.63 | 76.18 | 59.91 |
| KgCoOp | 71.20 | 64.10 | 48.97 | 50.69 | 76.70 | 60.12 |
| MaPLe | 70.72 | 64.07 | 49.15 | 50.90 | 76.98 | 60.27 |
| TCP | 70.92 | 64.42 | 49.33 | 50.78 | 77.11 | 60.41 |
| PSRC | 71.27 | 64.35 | 49.55 | 50.90 | **77.80** | 60.65 |
| **Ours** | 71.88 | 65.12 | **49.98** | **51.68** | 77.52 | **61.07** |

## 4.5 ABLATION STUDIES

**Contributions of major algorithm components.** From Table 5, we can see that both components contribute significantly to the enhanced performance. Among them, text-guided visual prompt tokens brings the largest performance improvement, for example, a notable 4.43% improvement in Novel. And the inclusion of the text-guided **CLS** token can further enhance the overall performance, demonstrating that multi-modal interaction should not be restricted to the certain prompt tokens. The baseline here is IVLP(Independent Vision Language Prompt), which contains independent prompts in vision and language branches.

Table 5: Effectiveness of different components in our method.

| Method | Base | Novel | HM |
|---|---|---|---|
| IVLP | 83.47 | 72.46 | 77.57 |
| + **CLS**-tg | 84.26 | 74.98 | 79.34 |
| + **VP**-tg | 84.88 | 76.89 | 80.69 |
| + **CLS**-tg + **VP**-tg | **85.10** | **77.73** | **81.24** |

Table 6: Comparison with other LLM-based methods

| Method | Base | Novel | HM |
|---|---|---|---|
| CoPrompt | 84.00 | 77.23 | 80.84 |
| LLaMP | 85.16 | 77.71 | 81.27 |
| CGP | 84.38 | 78.03 | 81.08 |
| ArGue | 83.77 | **78.74** | 81.18 |
| **Ours** | **85.57** | 78.42 | **82.35** |

**Compared to methods with help of LLM.** Some existing methods, such as CoPromp (Roy & Etemad, 2024), LLaMP (Chiang et al., 2024), CGP (Zhang et al., 2024) and ArGue (Tian et al., 2024), focus on harnessing the power of large language models (LLMs) to provide richer and more targeted textual knowledge for prompt construction. By simply replicating the process used by ArGue to generate more accurate and comprehensive textual knowledge through LLMs, the results in Table 6 demonstrate that our method can more effectively harness LLM-generated knowledge, leading to superior performance in HM.

**The Number $K$ of Most Relevant Textual Guidance Selected.** We examine the influence of varying the number of selected relevant classes ($K$) on performance, with the results presented in Table 9. Our findings indicate that the optimal performance is achieved when $K = 5$, and the results demonstrate that our method remains robust to variations in $K$, further underscoring the

Table 7: Number top-$K$ of most relevant text classes selected.

| Value of $K$ | 1 | 3 | **5** | 8 | 10 |
|---|---|---|---|---|---|
| **Base** | 84.52 | 84.88 | **85.10** | 84.86 | 84.94 |
| **Novel** | 76.84 | 77.35 | **77.73** | 77.58 | 77.42 |
| **HM** | 80.49 | 80.94 | **81.24** | 81.05 | 81.06 |

effectiveness of the proposed TKG module. Note that if $K$ is larger than the number of current classes or we use visual prompt token as queries, TKG will directly select textual guidance of all classes.

## 5 CONCLUSION

In this work, we present a novel approach called Text-Guided Visual Prompt Tuning (TGVP) for vision-language models. By leveraging the robust generalizability of textual knowledge, our method guides the generation of visual prompts through a unique Text-Knowledge Guidance Module. This

module dynamically integrates task-relevant textual knowledge into visual prompts using a cross-attention mechanism. The resulting text-guided visual prompts enhance the visual encoder's semantic awareness, thereby improving both generalization and discriminability across various scenarios. Extensive experiments demonstrate that TGVP not only significantly outperforms existing methods in base-to-novel generalization, cross-dataset transfer, and domain generalization tasks but also offers a substantial improvement in the adaptation of vision-language models.

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

# A APPENDIX

## A.1 MORE IMPLEMENTATION DETAILS

In addition to the implementation details mentioned in the main text, our code is based on Prompt-SRC's code, adhering to its deep prompt configuration, where deep text/visual prompts are set in the first 9 layer with our proposed Text-knowledge Guidance Module. Additionally, during the deployment of the TKG module, the textual guidance is applied to the visual prompt tokens in the first nine layers and the CLS token in the ninth layer. Finally, during training, we employed the standard cross-entropy loss and textual diversity loss as used in PromptSRC.

## A.2 MORE ABLATION STUDIES

**Selecting outputs from *Which* layers of the text encoder as textual knowledge source.** Since we have pointed that text embeddings can be leveraged as knowledge source at the cross-modal interaction, we evaluate the impact of using outputs from different layers of text encoder as the source of knowledge on performance. As shown in Table 9, our findings indicate that the optimal performance is achieved when $J = 12$, which demonstrates that the text embedding from the final layer contains the richest semantic information, making it the most suitable source of textual knowledge.

| $J$-th Layer | 9 | 10 | 11 | **12** |
|:---:|:---:|:---:|:---:|:---:|
| **Base** | 83.88 | 84.19 | 84.55 | **85.10** |
| **Novel** | 75.23 | 75.55 | 76.52 | **77.73** |
| **HM** | 79.31 | 79.63 | 80.33 | **81.24** |

Table 8: Using outputs from different layers of text encoder as the source of textual knowledge.

**Selection of hyper parameter $\lambda$** For the $\lambda$ parameter in the EMA mechanism, we uniformly set it to 0.5 across all experiments. To provide a comprehensive analysis, we also conducted additional ablation studies discussing the impact of different $\lambda$ values. The results demonstrate that as the parameter $\lambda$ increases, the strength of textual knowledge guidance intensifies, leading to a significant improvement in the model's generalization performance on novel classes. However, retaining a portion of the original visual prompt token information proves beneficial for enhancing the model's overall performance across both base and novel classes. To balance the model's performance on base and novel categories, we selected $\lambda$=0.5 as the optimal value.

| $\lambda$ | 0.1 | 0.3 | 0.5 | 0.8 | 1 |
|:---:|:---:|:---:|:---:|:---:|:---:|
| **Base** | 84.54 | 84.69 | **85.10** | 84.23 | 83.88 |
| **Novel** | 75.43 | 76.52 | **77.73** | 77.59 | 77.63 |
| **HM** | 79.73 | 80.40 | **81.24** | 80.77 | 80.63 |

Table 9: Abalation study of value $\lambda$

**Visualization** To showcase the discriminative power of the proposed TGVP in generating vision classifiers for prediction, we visualize the prediction probabilities across both base and novel classes. As illustrated in Figure 3, our TGVP achieves more pronounced inter-class separation than existing methods in both seen and unseen scenarios, highlighting its superior discriminability and generalization capacity

**Detailed experiment to show the results of Figure 1** To further substantiate the motivation presented in Figure 1, we conducted more detailed experiments across additional datasets. The results demonstrate that the generalization performance of visual prompts is weaker than that of text prompts, with the gap becoming more pronounced as the dataset difficulty increases. Additionally, the table also shows the performance of visual prompts after incorporating our proposed TGVP. It

Figure 3: Visualization of the prediction probability obtained by PromptSRC and Ours.

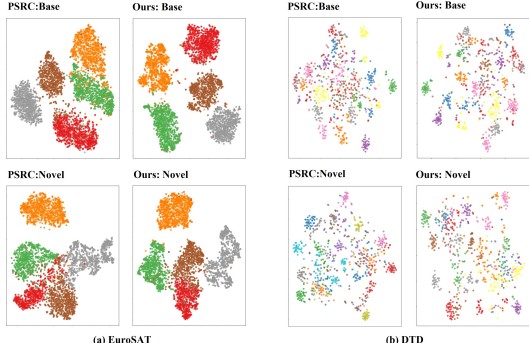

| Dataset | | Textual Prompt | Visual Prompt | Visual Prompt+TGVP |
|---|---|---|---|---|
| ImageNet | Base Acc. | 75.23 | 76.53 | 76.97 |
| | Novel Acc. | 65.67 | 63.77 | 66.36 |
| OxfordPets | Base Acc. | 94.68 | 95.59 | 95.77 |
| | Novel Acc. | 97.83 | 97.48 | 98.12 |
| FGVCAircraft | Base Acc. | 35.60 | 36.36 | 39.86 |
| | Novel Acc. | 27.96 | 22.26 | 36.89 |
| DTD | Base Acc. | 82.26 | 82.26 | 82.59 |
| | Novel Acc. | 56.64 | 51.68 | 60.14 |
| EuroSAT | Base Acc. | 91.31 | 94.88 | 97.23 |
| | Novel Acc. | 72.46 | 62.18 | 74.01 |
| Average over 11 datasets | Base Acc. | 82.89 | 83.13 | 84.16 |
| | Novel Acc. | 70.79 | 69.38 | 71.94 |

Table 10: Detailed performance comparison on individual datasets for showing effect

can be observed that the generalization performance of visual prompts improves significantly, even surpassing the generalization performance of textual prompts.

