# OpenReview forum: "Text-Guided Visual Prompt Tuning for Vision-Language Models"
_ICLR.cc/2025/Conference — Submitted to ICLR 2025_

### Official Review · Reviewer_8TFe · 2024-11-02

**Soundness:** 2
**Presentation:** 3
**Contribution:** 2
**Rating:** 5
**Confidence:** 4

**Summary:**

This article first analyzes the impact of unimodal prompts on the base-to-novel classification. It concludes that using text prompt representations to guide image prompt representations is beneficial. Therefore, the author proposes the TGVP method, which utilizes parameter-free cross-attention to guide the optimization of image prompt representations. The author demonstrates the effectiveness of this method through extensive experiments.

**Strengths:**

1. The presentation in this paper is quite clear, allowing for a relatively clear understanding of the author's methodological process.

2. The experimental content in this paper is comprehensive, effectively demonstrating the efficacy of the proposed method.

**Weaknesses:**

1. The author introduces too many variables and formulas in the method introduction section, which may cause some difficulties in understanding the author's method.

2. The author's motivation is derived from the analysis of unimodal prompts, leading to the conclusion of using text to guide visual prompts. However, the visual prompts obtained through the author's method are also unimodal. Therefore, I believe it would be beneficial to add the performance of the new visual prompts in Figure 1 to demonstrate the effectiveness of the author's method.

3. Guiding the representation generation of visual prompts through text has already been applied in MaPLE. Additionally, the author's cross-attention calculation appears to be similar to the parameter-free attention in CALIP[1].

[1] CALIP: Zero-Shot Enhancement of CLIP with Parameter-free Attention

**Questions:**

1. Referring to weakness 2, I believe it would be beneficial to include the optimized visual features based on the author's method to demonstrate its effectiveness.

2. In the description of the method, it would be helpful to reduce the introduction of new variables and include pseudocode to aid in understanding the author's approach.

3. Personally, I think that given the lack of novelty in the method presented, the paper should reduce the length devoted to describing the method. Instead, it could analyze what causes the differences in generalization capabilities between visual and text unimodal prompts, or explore whether encoder-only VLMs can be extended to decoder-only VLMs. Such analyses would make the work more impactful.

---

> ### Author Response · Authors · 2024-11-25
>
> > **W1**. The author introduces too many variables and formulas in the  method introduction section, which may cause some difficulties in  understanding the author's method.
>
> Thank you for your valuable feedback. We acknowledge that the introduction of multiple variables and formulas in the method section might make the explanation dense and potentially challenging to follow. To address this, we will revise the section to improve clarity and accessibility by streamlining the presentation, providing intuitive explanations alongside key formulas. We hope these revisions will enhance the readability and comprehension of our method.
>
>
>
> > **W2**. The author's motivation is derived from the analysis of  unimodal prompts, leading to the conclusion of using text to guide visual  prompts. However, the visual prompts obtained through the author's method  are also unimodal. Therefore, I believe it would be beneficial to add the     performance of the new visual prompts in Figure 1 to demonstrate the  effectiveness of the author's method.
>
> We sincerely appreciate your insightful feedback. In response, we have presented the performance of the optimized visual prompt. The results (seen in response to **reviewer a1Dz Q2**) reveal that the text-guided visual prompt achieves remarkable improvements on both base and novel classes, underscoring the efficacy of our approach in significantly enhancing the generalization capabilities of visual prompts.
>
>
>
> > **W3**. Guiding the representation generation of visual prompts through   text has already been applied in MaPLE. Additionally, the author's  cross-attention calculation appears to be similar to the parameter-free  attention in CALIP[1].
> >
> > [1] CALIP: Zero-Shot Enhancement of CLIP with Parameter-free Attention
>
> Thank you very much for your feedback. The concerns you raised are addressed in the **common response** and my reply to **Reviewer a1Dz's Weakness 1**. We hope these explanations resolve your concerns.
>
>
>
>
>
> > **Q1**. Referring to weakness 2, I believe it would be beneficial to  include the optimized visual features 1 based on the author's method to demonstrate its effectiveness.
>
> The details of this response can be found in **W2**.
>
>
>
>
>
> > **Q2**. In the description of the method, it would be helpful to reduce  the introduction of new variables and include pseudocode to aid in understanding the author's approach.
>
> We greatly appreciate your thoughtful feedback and suggestion. In response, we will revise the method description to reduce the introduction of new variables and will include pseudocode in the supplementary materials to facilitate a clearer understanding of our approach. We believe this will enhance the comprehensibility and accessibility of our method.
>
>
>
>
>
> > **Q3**. Personally, I think that given the lack of novelty in the  method presented, the paper should reduce the length devoted to describing the method. Instead, it could analyze what causes the differences in     generalization capabilities between visual and text unimodal prompts, or explore whether encoder-only VLMs can be extended to decoder-only VLMs.  Such analyses would make the work more impactful.
>
> Thank you very much for your thoughtful comments and suggestions. I believe that the novelty of our method has already been highlighted in the common response and the responses to the previous questions. Furthermore, in our response to Reviewer a1Dz, we conducted a more detailed experimental analysis of the motivation behind our method. These additional experiments further validate that our approach significantly enhances the generalization performance of visual prompts.  Nevertheless, we greatly appreciate your perspective, and we agree that further exploration into the generalization capabilities of different modalities, as well as the potential for targeted prompt design in decoder-only VLMs, will be valuable. These areas will certainly be the focus of our future work to further advance and refine the method.

---

> > ### Comment · Reviewer_8TFe · 2024-11-28
> >
> > Thank you for your detailed response.
> >
> > Referring to the author's explanation of their contributions in the introduction, the first contribution, which pertains to the motivation, appears to be an incremental improvement over MaPLE. The second contribution, concerning the methodology itself, as highlighted by the author in their response, seems to be more of an incremental modification to CALIP.
> >
> > Therefore, I have decided to maintain my current score for now.

---

### Official Review · Reviewer_a1Dz · 2024-11-03

**Soundness:** 2
**Presentation:** 3
**Contribution:** 2
**Rating:** 5
**Confidence:** 4

**Summary:**

This paper  proposes a method, named Text-Guided Visual Prompt Tuning (TGVP), to uniquely leverage the robust generalizability of textual knowledge to guide the generation of visual prompt.

**Strengths:**

-This paper emphasizes that high-level textual semantics are key to facilitating the learning of generalizable visual prompts

-Experiments show that the proposed method performs better than other existing approaches in base-to-novel generalization, cross-dataset transfer, and domain generalization tasks.

**Weaknesses:**

Cross-modallity attention has been done in CALIP, and the projection from text prompt to visual prompt in Figure 2 has also been done in MaPLe, which looks a bit like incremental;

[1] CALIP: Zero Shot Enhancement of CLIP with Parameter free Attention (AAAI 2023)

**Questions:**

-Regarding cross-modalality attention on Idea, CALIP: Zero Shot Enhancement of CLIP with Parameter free Attention (AAAI 2023) has already been done (very similar); This paper lacks this reference;

-This paper uses Figure 1 to express his motivation, aiming to demonstrate that the generalization ability of visual prompts is not as good as that of text prompts; However, only two small datasets, Eurosat (remote sensing) and DTD (texture), were displayed. Both datasets are small and very fine-grained. It is interensting that do similar experiments on large dataset, such as ImageNet, or the mean of all 11 datasets;

-In some experimental implementation details, such as line 273, the setting of the number of layers for the visual prompt and the comparative experiment on the number of layers are missing; 260 line EMA method, lacking setting of λ hyperparameter;

---

> ### Author Response · Authors · 2024-11-25
>
> > **W1:** Cross-modallity attention has been done in CALIP, and the projection from text prompt to visual prompt in Figure 2 has also been done in MaPLe, which looks a bit like incremental;
> >
> > [1] CALIP: Zero Shot Enhancement of CLIP with Parameter free Attention (AAAI 2023)
>
>
>
> The distinctions between our method and approaches such as MaPLe have been elaborated in detail in the common response. The primary innovations of our method are as follows:  our method introduces several targeted improvements to **enable more comprehensive cross-modal interaction** and **improved generalization to unseen categories,** which constitute the core contributions of our work:
>
> 1. **Text Embedding as a Cross-Modal Information Source**: For the first time, we propose leveraging the text embeddings—output from the text encoder and rich in high-level semantics—as the textual information source for cross-modal interaction. This ensures a more comprehensive and semantically robust exchange of information.
> 2. **Text-Knowledge Guidance Module**: We propose a novel Text-Knowledge Guidance Module, which can dynamically transfer textual knowledge to guide the generation of visual prompts. This makes the visual prompts semantically aware and adaptable to both seen and unseen classes, thereby enhancing the generalization capability of the model.
>
> While our method may share some superficial similarities with the cross-attention design in CALIP, there are significant differences in terms of design objectives, methodology, and implementation details:
>
> **1. Difference in Objectives**
>
> - **CALIP** aims to achieve bidirectional feature enhancement between textual and  visual features through a parameter-free cross-attention mechanism. In contrast, **our method** focuses on selecting the most relevant text  category knowledge as guidance for a specific vision task. This demonstrates a significant difference in the **functional objectives**:
>   - **CALIP** emphasizes feature complementarity and  bidirectional fusion.
>   - **Our method** is task-driven, prioritizing **task-specific  relevance**
>
> **2. Key Design Differences**
>
> - **CALIP**  employs a fully parameter-free attention mechanism, directly applying  SoftMax to $F_t$ and $F_v$ .In contrast, **our method** incorporates several critical steps:
>   - **Top-k Selection:** Our method computes a similarity matrix and  selects the top-k most relevant text category tokens for each visual prompt. This selection process does not exist in CALIP.
>   - **Temperature Modulation Mechanism:** Our method uses a  temperature parameter τ\tauτ to control the sharpness of the similarity  distribution, enhancing task adaptability. CALIP does not include such a mechanism.
>   - **Weighted Feature Aggregation:** Our method aggregates   top-k guidance using attention weights, producing new task-specific text guidance. **CALIP does not involve such a selection and aggregation process**
>
> **3. Highlighting Innovations**
>
> - The unique aspect of **our method** lies in its **"most relevant text category selection"** strategy, which is crucial for solving multimodal fusion challenges in specific vision tasks:
>   1. **Explicit Guidance Selection:** Instead of  indiscriminately fusing textual and visual features, our method focuses  on selecting task-relevant text categories through top-k filtering.
>   2. **Hierarchical Task Guidance:** Our method emphasizes  the selection of specific guidance for different levels of vision tasks,  whereas CALIP's attention mechanism lacks such hierarchical design.
>
>
>
>
>
>
>
> > **Q1**：-Regarding cross-modalality attention on Idea, CALIP: Zero Shot Enhancement of CLIP with Parameter free Attention (AAAI 2023) has already been done (very similar); This paper lacks this reference;
>
> The details of this response can be found in **W1**.

---

> ### Author Response · Authors · 2024-11-25
>
> > **Q2**：-This paper uses Figure 1 to express his motivation, aiming to demonstrate that the generalization ability of visual prompts is not as good as that of text prompts; However, only two small datasets, Eurosat (remote sensing) and DTD (texture), were displayed. Both datasets are small and very fine-grained. It is interensting that do similar experiments on large dataset, such as ImageNet, or the mean of all 11 datasets;
>
> Thank you very much for your suggestion. Based on your feedback, we conducted more detailed experiments, and the results are presented in the table below.
>
> The results demonstrate that the generalization performance of visual prompts is weaker than that of text prompts, with the gap becoming more pronounced as the dataset difficulty increases. Additionally, the table also shows the performance of visual prompts after incorporating our proposed TGVP. It can be observed that the generalization performance of visual prompts improves significantly, even surpassing the generalization performance of textual prompts.
>
> |         **Dataset**          |                | **Textual Prompt** | **Visual Prompt** | **Visual Prompt + TGVP** |
> | :--------------------------: | -------------- | ------------------ | ----------------- | ------------------------ |
> |         **ImageNet**         | **Base Acc.**  | 75.23              | 76.53             | **76.97**                |
> |                              | **Novel Acc.** | 65.67              | 63.77             | **66.36**                |
> |        **OxfordPets**        | **Base Acc.**  | 94.68              | 95.59             | **95.77**                |
> |                              | **Novel Acc.** | 97.83              | 97.48             | **98.12**                |
> |       **FGVCAircraft**       | **Base Acc.**  | 35.60              | 36.36             | **39.86**                |
> |                              | **Novel Acc.** | 27.96              | 25.26             | **36.89**                |
> |           **DTD**            | **Base Acc.**  | 82.26              | 82.26             | **82.59**                |
> |                              | **Novel Acc.** | 56.64              | 51.68             | **60.14**                |
> |         **EuroSAT**          | **Base Acc.**  | 91.31              | 94.88             | **97.23**                |
> |                              | **Novel Acc.** | 72.46              | 62.18             | **74.01**                |
> | **Average over 11 datasets** | **Base Acc.**  | 82.89              | 83.13             | **84.16**                |
> |                              | **Novel Acc.** | 70.79              | 69.38             | **71.94**                |
>
>
>
>
>
> > **Q3**: -In some experimental implementation details, such as line 273, the setting of the number of layers for the visual prompt and the comparative experiment on the number of layers are missing; 260 line EMA method, lacking setting of λ hyper parameter;
>
> In our work, the number of layers for both visual and textual prompts is consistently set to the first 9 layers. This configuration aligns with prior works, such as PromptSRC and MaPLe, which also adopt a similar "Deep Prompt" strategy.
>
> For the λ parameter in the EMA mechanism, we uniformly set it to 0.5 across all experiments.
>
> To provide a comprehensive analysis, we also conducted additional ablation studies discussing the impact of different λ values. The results demonstrate that as the parameter λ increases, the strength of textual knowledge guidance intensifies, leading to a significant improvement in the model's generalization performance on novel classes. However, retaining a portion of the original visual prompt token information proves beneficial for enhancing the model's overall performance across both base and novel classes. To balance the model's performance on base and novel categories, we selected λ=0.5 as the optimal value.
>
> | **λ**     | 0.1   | 0.3   | 0.5       | 0.8   | 1     |
> | --------- | ----- | ----- | --------- | ----- | ----- |
> | **Base**  | 84.54 | 84.69 | **85.10** | 84.23 | 83.88 |
> | **Novel** | 75.43 | 76.52 | **77.73** | 77.59 | 77.63 |
> | **HM**    | 79.73 | 80.40 | **81.24** | 80.77 | 80.63 |

---

> > ### Comment · Reviewer_a1Dz · 2024-12-03
> > **Official Comment by Reviewer a1Dz**
> >
> > Thank you for your response.
> >
> > After reading the rebuttal and other reviewers' comments, the contribution is a incremental modification to CALIP and MaPLE. Thus, I have decided to keep my score.

---

### Official Review · Reviewer_3h8Y · 2024-11-03

**Soundness:** 3
**Presentation:** 3
**Contribution:** 2
**Rating:** 5
**Confidence:** 3

**Summary:**

The paper proposed a novel prompt-tuning method for VLMs. At its core, the proposed method uses visual prompt tokens and the CLS token to attend to the text prompts and get "text guidance" which is subsequently added (through moving average) to the visual prompts to get "text-guided visual prompts".

**Strengths:**

The paper is well-organized and is easy to read. Experiments includes 4 set of experiments on 11 dataset which is good.

**Weaknesses:**

The paper seems very similar to MaPLe (Khattak 2023a) and PromptSRC (kHATTAK 2023B) in that they all jointly learn visual and/or text prompts. The related work section briefly mentions them but does not really discuss them adequately.

In the base-to-novel generalization experiment (table 1) the average improvement under the HM column (harmonic mean of base and novel classes) is 1.27% over 11 dataset. However, a closer look reveals that this improvement is mostly due the EuroSAT dataset which shows 8% improvement . Excluding that dataset, the average improvement over the remaining 10 dataset is only 0.35% which is a very marginal improvement .

In Table 4 about the domain generalization, the TCP method is missing. Considering that TCP seems to be among top performing methods in other experiments (Table 1-3), including the results of TCP in table 4 will be helpful.

**Questions:**

See my comments above.
Also, the proposed method shows a strong performance on the EuroSAT dataset across various experiments. Performance on the other 10 datsets are relatively much lower. A discussion on what is special about the EuroSAT dataset would be insightful.

---

> ### Author Response · Authors · 2024-11-25
>
> We sincerely appreciate your valuable comments and suggestions. Below, we provide detailed, point-by-point responses to address your concerns. We hope these replies effectively resolve the issues you have raised.
>
>
>
> > **W1**：The paper seems very similar to MaPLe (Khattak 2023a) and PromptSRC (kHATTAK 2023B) in that they all jointly learn visual and/or text prompts. The related work section briefly mentions them but does not really discuss them adequately.
>
> We have elaborated on this point in detail in the common review section and hope it addresses your concerns effectively.
>
>
>
> > **W2**：In the base-to-novel generalization experiment (table 1) the average improvement under the HM column (harmonic mean of base and novel classes) is 1.27% over 11 dataset. However, a closer look reveals that this improvement is mostly due the EuroSAT dataset which shows 8% improvement . Excluding that dataset, the average improvement over the remaining 10 dataset is only 0.35% which is a very marginal improvement .
>
>
>
> Regarding the notable performance improvement of our method on the EuroSAT dataset, compared to other image classification datasets discussed in the paper, we identify the following key characteristics of EuroSAT: it contains a limited number of categories (only 10) and poses a greater challenge for zero-shot recognition by pre-trained models (e.g., the CLIP zero-shot classification accuracy is relatively low). We attribute these challenges to the nature of EuroSAT as a dataset of satellite remote sensing images, **which are of low resolution (32×32) and exhibit a substantial domain gap from natural images**. **Many images consist of pure color blocks that are difficult to discern even for humans, further complicating the task for pre-trained models.**
>
> Our method addresses these challenges effectively by dynamically injecting text-based category knowledge into the visual prompt, thereby enhancing the intra-class compactness and inter-class separability of the visual features generated by the visual encoder. The effectiveness of this approach is further illustrated in our latest visualizations.
>
>
>
> > **W3**：In Table 4 about the domain generalization, the TCP method is missing. Considering that TCP seems to be among top performing methods in other experiments (Table 1-3), including the results of TCP in table 4 will be helpful.
>
>
>
> Thank you for your suggestion. We have reproduced TCP and included its results in the domain generalization experiments.
>
> |             | **Source**   | **Target** |             |           |           |           |
> | ----------- | ------------ | ---------- | ----------- | --------- | --------- | --------- |
> |             | **ImageNet** | **-V2**    | **-Sketch** | **-A**    | **-R**    | **Avg.**  |
> | **CLIP**    | 66.73        | 60.83      | 46.15       | 47.77     | 73.96     | 57.18     |
> | **WiSE-FT** | **73.02**    | **65.19**  | 49.09       | 49.81     | 77.63     | 60.43     |
> | **CoOp**    | 71.51        | 64.20      | 47.99       | 49.71     | 75.21     | 59.28     |
> | **CoCoOp**  | 71.02        | 64.07      | 48.75       | 50.63     | 76.18     | 59.91     |
> | **KgCoOp**  | 71.20        | 64.10      | 48.97       | 50.69     | 76.70     | 60.12     |
> | **MaPLe**   | 70.72        | 64.07      | 49.15       | 50.90     | 76.98     | 60.27     |
> | **TCP**     | 70.92        | 64.42      | 49.33       | 50.78     | 77.11     | 60.41     |
> | **PSRC**    | 71.27        | 64.35      | 49.55       | 50.90     | **77.80** | 60.65     |
> | **Ours**    | 71.88        | 65.12      | **49.98**   | **51.68** | 77.52     | **61.07** |
>
>
>
> > **Q1**：See my comments above. Also, the proposed method shows a strong performance on the EuroSAT dataset across various experiments. Performance on the other 10 datsets are relatively much lower. A discussion on what is special about the EuroSAT dataset would be insightful.
>
> The details of this response can be found in **W2*.

---

> > ### Comment · Reviewer_3h8Y · 2024-12-03
> >
> > I went over the responses. My main concern is the incremental contributions, as noted by other reviewers. Also note that the performance improvements excluding the EuroSAT dataset is much less than the reported average performance over all datasets. I maintain my initial score.

---

### Official Review · Reviewer_zQCx · 2024-11-07

**Soundness:** 2
**Presentation:** 3
**Contribution:** 2
**Rating:** 5
**Confidence:** 5

**Summary:**

This paper introduces Text-Guided Visual Prompt Tuning (TGVP) to enhance the generalization of vision-language models (VLMs) for diverse downstream tasks. Traditional methods struggle to incorporate task-relevant textual knowledge into visual prompts, limiting their adaptability to novel classes. TGVP addresses this by using a Text-Knowledge Guidance Module with a cross-attention mechanism, allowing visual prompts to better capture semantic context. Experiments show TGVP significantly improves VLM performance in generalization, cross-dataset transfer, and domain adaptation tasks.

**Strengths:**

This paper addresses a compelling challenge in vision-language model adaptation: improving generalization to unseen tasks and classes. Existing prompt tuning methods often overlook the benefits of integrating textual knowledge into visual prompts.  By leveraging textual guidance, TGVP demonstrates superior generalization performance, particularly in base-to-novel class adaptation, cross-dataset transfer, and domain generalization, addressing common limitations in traditional prompt tuning methods.

**Weaknesses:**

W1:  The performance improvement demonstrated by the proposed method is relatively modest, limiting the practical impact and significance of the contribution. Further analysis or comparison with a broader range of baselines could help clarify the advantages and effectiveness of the approach.

W2: The paper lacks coverage of some important related works, particularly in areas that could provide a deeper contextual foundation for the proposed method. Including a more comprehensive review of relevant studies, especially recent advances in prompt tuning and ensemble learning, would enhance the paper's contribution and situate it more clearly within the broader research landscape.

**Questions:**

Q1: What is the function of the "project" component in your model? How would altering its structure impact performance?

Q2: Could the authors consider adding the ensemble baselines, such as the WiSE-FT method, to provide a more comprehensive comparison?
WiSE-FT: Robust fine-tuning of zero-shot models

Q3: Could the authors clarify the definition of "P" in Equation (7)? Additionally, could you explain the process for obtaining T^{topk}_{j} and I^{topk}_{j}?

---

> ### Author Response · Authors · 2024-11-25
>
> We sincerely appreciate your valuable comments and suggestions. Below, we provide detailed, point-by-point responses to address your concerns. We hope these replies effectively resolve the issues you have raised.
>
> > **Weakness 1**:
> Thank you for your valuable feedback and suggestions. I believe the novelty of our method has already been clearly outlined in the unified response. Regarding your point about the proposed method's performance being relatively modest, we would like to emphasize that the experimental setups in the field of prompt tuning are already quite challenging. In this context, performance improvements over existing methods are typically modest, often under 5%. However, our approach demonstrates consistent improvement across 11 different image classification tasks, with notable gains of approximately 8% on datasets that differ significantly from natural images, such as EuroSAT. Additionally, in Table 6 of the paper, we compared our method with state-of-the-art approaches that utilize LLMs, and the results show that our method achieves the best performance even when augmented with LLMs. Based on these observations, we believe that the improvements demonstrated by our method are significant within the context of the field.
>
> > **Weakness2**:
> We sincerely appreciate your insightful feedback and suggestions. As outlined in the experimental section, our method has been evaluated across a diverse array of standard experimental setups in the prompt tuning domain, and we have conducted comparisons with both the most recent and seminal methods in the field. Nonetheless, we acknowledge your valuable point regarding the inclusion of ensemble learning approaches. In response, we have expanded our evaluation to incorporate ensemble learning techniques as additional baselines. We believe that this enhancement will not only strengthen the foundation of our work but also position it more effectively within the broader research landscape.
>
> > **Q1**: What is the function of the "project" component in your model? How would altering its structure impact performance?
>
> In our work, the “project” component is designed to transfer text-embedding, which contains high-level semantic information, into vision prompt token space for further cross-modality interaction. The projector is constructed using a simple "Linear+ReLU+Linear" structure, with its primary structural variation determined by the dimensionality of the intermediate layer, $D_{\text{dim}}$. We conducted ablation studies on $D_{\text{dim}}$, and the results demonstrate that both excessively low and excessively high values for $D_{\text{dim}}$ negatively impact the model's final performance. Based on these findings, we selected $D_{\text{dim}} = 128$ as the optimal parameter.
> > **Q2**: Could the authors consider adding the ensemble baselines, such as the WiSE-FT method, to provide a more comprehensive comparison? WiSE-FT: Robust fine-tuning of zero-shot models
>
> Thank you for the insightful suggestion. We will incorporate WiSE-FT into the baseline in the experiment of domain-generalization to provide a more comprehensive and robust comparison in our revised submission.
> |             | **Source**   | **Target** |             |           |           |           |
> | ----------- | ------------ | ---------- | ----------- | --------- | --------- | --------- |
> |             | **ImageNet** | **-V2**    | **-Sketch** | **-A**    | **-R**    | **Avg.**  |
> | **CLIP**    | 66.73        | 60.83      | 46.15       | 47.77     | 73.96     | 57.18     |
> | **WiSE-FT** | **73.02**    | **65.19**  | 49.09       | 49.81     | 77.63     | 60.43     |
> | **CoOp**    | 71.51        | 64.20      | 47.99       | 49.71     | 75.21     | 59.28     |
> | **CoCoOp**  | 71.02        | 64.07      | 48.75       | 50.63     | 76.18     | 59.91     |
> | **KgCoOp**  | 71.20        | 64.10      | 48.97       | 50.69     | 76.70     | 60.12     |
> | **MaPLe**   | 70.72        | 64.07      | 49.15       | 50.90     | 76.98     | 60.27     |
> | **TCP**     | 70.92        | 64.42      | 49.33       | 50.78     | 77.11     | 60.41     |
> | **PSRC**    | 71.27        | 64.35      | 49.55       | 50.90     | **77.80** | 60.65     |
> | **Ours**    | 71.88        | 65.12      | **49.98**   | **51.68** | 77.52     | **61.07** |
>
> > **Q3**: Could the authors clarify the definition of "P" in Equation (7)? Additionally, could you explain the process for obtaining T^{topk}*{j} and I^{topk}*{j}?
>
> In Equation (7), "P" represents the visual prompt tokens. Regarding the selection process for $T_{topk}$, we compute the attention map between the visual prompt tokens and the text embeddings through dot product. From the $N_c$ category text embeddings, we select the $top_k$ categories most relevant to the visual prompt tokens based on this attention map. Subsequently, the text embeddings of these $top_k$ categories are utilized to inject textual information into the visual prompts.

---

> ### Author Response · Authors · 2024-11-29
> **Additional response to Weakness 1.**
>
> To further validate the motivation and effectiveness of our method, we conducted more extensive experiments across 11 downstream datasets. Below, we present some representative results. It is evident that the text-guided approach enables the visual prompt to achieve remarkable improvements on both base and novel classes, highlighting the efficacy of our method in significantly enhancing the generalization capabilities of visual prompts.
>
> Notably, in these experiments, no prompt was applied to the text encoder; the guiding textual information consisted solely of the text embeddings output by the original CLIP text encoder. This further underscores the exceptional performance of our method in optimizing the effectiveness of visual prompts.
>
> |         **Dataset**          |                | **Textual Prompt** | **Visual Prompt** | **Visual Prompt + TGVP** |
> | :--------------------------: | -------------- | ------------------ | ----------------- | ------------------------ |
> |         **ImageNet**         | **Base Acc.**  | 75.23              | 76.53             | **76.97**                |
> |                              | **Novel Acc.** | 65.67              | 63.77             | **66.36**                |
> |        **OxfordPets**        | **Base Acc.**  | 94.68              | 95.59             | **95.77**                |
> |                              | **Novel Acc.** | 97.83              | 97.48             | **98.12**                |
> |       **FGVCAircraft**       | **Base Acc.**  | 35.60              | 36.36             | **39.86**                |
> |                              | **Novel Acc.** | 27.96              | 25.26             | **36.89**                |
> |           **DTD**            | **Base Acc.**  | 82.26              | 82.26             | **82.59**                |
> |                              | **Novel Acc.** | 56.64              | 51.68             | **60.14**                |
> |         **EuroSAT**          | **Base Acc.**  | 91.31              | 94.88             | **97.23**                |
> |                              | **Novel Acc.** | 72.46              | 62.18             | **74.01**                |
> | **Average over 11 datasets** | **Base Acc.**  | 82.89              | 83.13             | **84.16**                |
> |                              | **Novel Acc.** | 70.79              | 69.38             | **71.94**                |

---

> > ### Comment · Reviewer_zQCx · 2024-11-30
> > **Official Comment by Reviewer zQCx**
> >
> > I completely agree with Reviewer 8TFe's opinion that the contributions of this paper are indeed incremental to previous work. Therefore, I will maintain my score.

---

### Author Response · Authors · 2024-11-25

We sincerely appreciate the reviewers' thorough and thoughtful feedback on our submission. To address the concerns effectively, we first provide a unified response to the common issues raised by multiple reviewers. Subsequently, we address each reviewer's specific comments in detail.

Firstly, to address the reviewers' concerns regarding the similarity of our method to MaPLe and PromptSRC, we begin by reviewing the main innovations of these two methods and highlighting their similarities and differences in relation to our approach.

MaPLe aims to **establish a mapping from textual prompts to visual prompts** for better alignment of two modalities.

PromptSRC aims using guides the prompts to optimize for both task-specific and task-agnostic general representations **using several novel regularizations**.

The key distinction between our approach with MaPLe and PromptSRC, lies in two aspects：the **prompt structure** and the **mechanism of cross-modal interaction**.

In terms of prompt structure：As illustrated in Figure 1(a), MaPLe confines the interaction between visual prompts and textual prompts to the **prompt token level**, while in PromptSRC, the two modalities **remain entirely independent**, **lacking cross-modal interactions**.

In our view, **cross-modal interaction limited to the prompt token** level has two fundamental limitations:

1. **The source of textual information is confined to fixed text prompts**, which are uniform across both seen and unseen scenarios, thereby hindering effective adaptation to unseen classes.
2. **the simple symmetrically projection mechanism is insufficient for information interaction between visual and textual modalities**, as textual features naturally contain semantic information while visual features carry local patch information from the current image.

To address the aforementioned limitations, our method introduces several targeted improvements to **enable more comprehensive cross-modal interaction** and **improved generalization to unseen categories,** which constitute the core contributions of our work:

1. **Text Embedding as a Cross-Modal Information Source**: For the first time, we propose leveraging the text embeddings—output from the text encoder and rich in high-level semantics—as the textual information source for cross-modal interaction. This ensures a more comprehensive and semantically robust exchange of information.
2. **Text-Knowledge Guidance Module**: We propose a novel Text-Knowledge Guidance Module, which can dynamically transfer textual knowledge to guide the generation of visual prompts. This makes the visual prompts semantically aware and adaptable to both seen and unseen classes, thereby enhancing the generalization capability of the model.

We hope that the above explanation provides a more comprehensive understanding of the motivation and  innovation embodied in our proposed method.

Below, we provide detailed, point-by-point responses to address your concerns. We hope these replies effectively resolve the issues you have raised.

---

### Meta-Review · Area_Chair_Ybvm · 2024-12-19

**Metareview:**

This paper focused on prompt tuning vision-language models (VLMs) to downstream tasks for generalization. This work argued that unseen class generalization remains challenging as visual prompt is hard to capture task-relevant textual knowledge. To tackle this challenge, this work proposed text-guided visual prompt tuning that leverages textural knowledge to guide the generation of visual prompt.

Main Strengths: (1) The presentation of the paper is clear. (2) The experiments are comprehensive to validate the effectiveness of the methods, covering multiple benchmarks and settings.

Main Weaknesses: (1) The improvement on 10 datasets is marginal except for EuroSAT, which challenges the effectiveness of the proposed method. (2) The novelty compared to MaPLe, CALIP, and PromptSRC is incremental.

This paper received four borderline negative ratings, i.e., 5, 5, 5, 5. The main reasons for rejecting the paper are the marginal improvement on 10 datasets and the novelty compared to MaPLe and CALIP. The AC does not have strong reasons to overturn reviewers' recommendations, and encourage the authors to include the discussion details in the future version to clarify the novelty more clearly.

**Additional Comments On Reviewer Discussion:**

After discussion, reviewers still have concerns about marginal improvement and novelty compared to related works, which are the main reasons for rejection.

---

### Decision · Program_Chairs · 2025-01-22

Reject